Characterization of viral pathogens associated with symptomatic upper respiratory tract infection in adults during a low COVID-19 transmission period

http://orcid.org/0000-0003-1814-2798 Sandybayev Nurlan 1 nurlan.s@kaznaru.edu.kz
Beloussov Vyacheslav 1 2
http://orcid.org/0000-0003-3399-2942 Strochkov Vitaliy 1
http://orcid.org/0000-0003-4219-1055 Solomadin Maxim 3
Granica Joanna 2
http://orcid.org/0000-0002-7136-7921 Yegorov Sergey 4 5 yegorovs@mcmaster.ca
1 Kazakhstan-Japan Innovation Center, Kazakh National Agrarian Research University , Almaty , Kazakhstan
2 TreeGene Molecular Genetics Laboratory , Almaty , Kazakhstan
3 School of Pharmacy, Karaganda Medical University , Karaganda , Kazakhstan
4 Michael G. DeGroote Institute for Infectious Disease Research; McMaster Immunology Research Centre; Department of Biochemistry and Biomedical Sciences, McMaster University , Hamilton , Canada
5 School of Sciences and Humanities, Nazarbayev University , Astana , Kazakhstan
Uversky Vladimir
Electronic publication date: 2023 Mar 13
Publication date: 2023
Volume: 11
Electronic Location ID: e15008
Received 2022 Nov 25; Accepted 2023 Feb 15
Copyright: © 2023 Sandybayev et al.
Copyright year: 2023
Copyright holder: Sandybayev et al.
License: This is an open access article distributed under the terms of the Creative Commons Attribution License, which permits unrestricted use, distribution, reproduction and adaptation in any medium and for any purpose provided that it is properly attributed. For attribution, the original author(s), title, publication source (PeerJ) and either DOI or URL of the article must be cited.
License URL: https://creativecommons.org/licenses/by/4.0/

Keywords: Respiratory tract infections, COVID-19, SARS-CoV-2, Human parainfluenza, Rhinovirus, PCR, Central Asia, NGS, Influenza, Rhinovirus

Funding: Science Committee of the Ministry of Education and Science of the Republic of Kazakhstan AP09259192 This work was supported by the Science Committee of the Ministry of Education and Science of the Republic of Kazakhstan, Grant No. AP09259192. The funders had no role in study design, data collection and analysis, decision to publish, or preparation of the manuscript.

==============================
Background

The epidemiology of respiratory tract infections (RTI) has dramatically changed over the course of the COVID-19 pandemic. A major effort in the clinical management of RTI has been directed toward diagnosing COVID-19, while the causes of other, common community RTI often remain enigmatic. To shed light on the etiological causes of RTI during a low COVID-19 transmission period in 2021, we did a pilot study using molecular testing for virologic causes of upper RTI among adults with respiratory symptoms from Almaty, Kazakhstan.

Methods

Adults presenting at two public hospitals with respiratory symptoms were screened using SARS-CoV-2 PCR on nasopharyngeal swabs. A subset of RTI+, COVID-19-negative adults (n = 50) was then tested for the presence of common RTI viruses and influenza A virus (IAV). Next generation virome sequencing was used to further characterize the PCR-detected RTI pathogens.

Results

Of 1,812 symptomatic adults, 21 (1.2%) tested SARS-CoV-2-positive. Within the COVID-19 negative outpatient subset, 33/50 subjects (66%) had a positive PCR result for a common community RTI virus, consisting of human parainfluenza virus 3-4 (hPIV 3-4) in 25/50 (50%), rhinovirus (hRV) in 2 (4%), hPIV4-hRV co-infection in four (8%) and adenovirus or the OCR43/HKU-1 coronavirus in two (4%) cases; no IAV was detected. Virome sequencing allowed to reconstruct sequences of most PCR-identified rhinoviruses and hPIV-3/human respirovirus-3.

Conclusions

COVID-19 was cause to a low proportion of symptomatic RTI among adults. Among COVID-negative participants, symptomatic RTI was predominantly associated with hPIV and hRV. Therefore, respiratory viruses other than SARS-CoV-2 should be considered in the clinical management and prevention of adult RTI in the post-pandemic era.

Background

While the COVID-19 pandemic has exerted an enormous impact on the economy and healthcare globally, it has also altered the epidemiology of other, common community respiratory tract infections (RTI) (Chow, Uyeki & Chu, 2022). Thus, the transmission of some RTI, such as influenza and respiratory syncitial virus (RSV) has diminished (Wan et al., 2021; Olsen et al., 2021), while the incidence of other RTI, such as rhinovirus (RV) infections has risen, exceeding SARS-CoV-2-caused morbidity in some populations (Varela et al., 2022; Hyams et al., 2022).

In the past, clinical algorithms for managing non-severe RTI seldom involved diagnostic testing to identify specific causes of infection, largely due to an unclear benefit-to-cost ratio of molecular testing (Mahony et al., 2009). The recent availability of SARS-CoV-2 PCR and ELISA-based rapid testing has allowed differentiation of RTI with underlying “COVID-19” vs. “non-COVID” aetiology.

The global RTI landscape has been rapidly evolving, owing to a combination of growing natural and vaccine-elicited COVID-19 immunity and re-emergence of seasonal community circulating viruses. Given that SARS-CoV-2 infection has recently been cause to only a small portion of all RTI (Chow, Uyeki & Chu, 2022; Olsen et al., 2021; Varela et al., 2022), it has become clear that a better understanding of regional RTI dynamics is vital to ensure effective healthcare resource allocation and preparedness to mitigate any emerging infection threats. The latter is especially critical for resource-limited regions such as Central Asia, where governments have struggled to meet the public health demands raised by the COVID-19 pandemic (Yegorov et al., 2021; Balakrishnan, 2020).

Kazakhstan began COVID-19 screening in early 2020, the first among the Central Asian states, with a country-wide emergency state declared in March 2020 (Yegorov et al., 2021). By spring-summer of 2021 (the period when participant recruitment was done in the current study), up to 60% of population had been exposed to SARS-CoV-2 (Kadyrova et al., 2022; Smagul et al., 2022) and, while COVID-19 rates appeared to decrease during this period, national surveillance systems reported a rise in symptomatic RTI of non-COVID aetiology (Coronavirus2020, 2023). Kazakhstan’s climate and seasonality are similar to those of other Northern Hemisphere countries with continental climates. Although data on the seasonal community RTI transmission from Kazakhstan are limited, pre-pandemic studies from neighbouring Russia suggest that the regional seasonal RTI and influenza followed the patterns seen in other Northern Hemisphere regions (Moriyama, Hugentobler & Iwasaki, 2020), whereby RSV, hCoV and influenza tend to peak in late autumn through spring (~November–April), HRV is most prevalent in spring (~March–May) and autumn (~August–October), while parainfluenza circulation is strain-specific, spanning summer and autumn (Jun–Nov)(Sergeeva et al., 2013; Kurskaya et al., 2022; Tatochenko et al., 2010; Tabyshova et al., 2020).

Given the urgent need to gain insight into the regional RTI dynamics, here we aimed to collect pilot data in the largest city and former capital of Kazakhstan during a period of low COVID-19 transmission in the spring of 2021. Therefore, we performed PCR screening for SARS-CoV-2, influenza A virus (IAV) and common community RTI viruses among adults presenting with upper respiratory tract symptoms at two public hospitals in Almaty, and subsequently investigated the diversity of PCR-identified viruses using next generation sequencing.

Materials and Methods

Study setting and participant recruitment

This study was conducted at two public outpatient clinics, city polyclinics #5 and #36, in Almaty, the capital of Almaty region in South-Eastern Kazakhstan (Fig 1). Participants were screened in May–June 2021. Adults (aged 17–45) presenting with RTI symptoms, such as cough, sore throat, runny nose and/or fever, to the hospital outpatient departments were invited to participate in the study. At the time of screening, samples were collected for clinical laboratory testing of COVID-19 and the outpatients were invited to participate in the study by donating an additional nasopharyngeal swab for research purposes. Participants were recruited based on the availability of consent and willingness to have one nasopharyngeal swab collected for research purposes, in addition to any clinical samples acquired by clinic staff for diagnostic purposes. When the results of in-clinic COVID-19 diagnostic testing became available, samples from 50 COVID-19-negative participants were used for repeat SARS-CoV-2 and multiplex RTI PCR done in the research lab. The sample size of this pilot study was determined based on the available study budget.

Figure 1 Map of study location.

Images sourced from: GISGeography (https://gisgeography.com/kazakhstan-map/), World-Grain.com (https://www.world-grain.com/articles/13838-focus-on-kazakhstan), Wikimedia (https://commons.wikimedia.org/wiki/File:Almaty_districts.svg).

Ethics approval and consent to participate

All study procedures were approved by the Commission on Bioethics of KazNARU (dated October 15, 2020). Written informed consent was obtained from all participants.

Sample collection and processing

Nasopharyngeal swabs were collected following the national guidelines using sampling swabs (Remel R12504; Thermo Fisher Scientific, Waltham, MA, USA) into Microtest M4-RT (cat#R12505) tubes containing 3 ml of media optimized for virologic procedures (Thermo Fisher Scientific, Waltham, MA, USA). Samples were transported to the laboratory on ice and stored at −80 °C prior to analyses. Swabs were removed from tubes, and the sample media were partitioned such that 200 µl of sample was used for PCR, while the rest (~2.7 ml) of sample was used for viral purification and enrichment. For the PCR assays, samples were directly used for nucleic acid extraction, while for the sequencing library preparation samples underwent virus enrichment prior to DNA/RNA extraction.

Nucleic acid extraction

Viral nucleic acids were isolated from samples by the MagMAX™ Total kit Nucleic acid Isolation Kit (cat#M1840), Thermo Fisher Scientific (Waltham, MA, USA), in accordance with the manufacturer’s instructions. The quality of isolated nucleic acids was assessed on a NanoDrop 2000 spectrophotometer (Thermo Fisher Scientific, Waltham, MA, USA) and fluorometer (Qubit 4″ and Qubit™ RNA HS Assay Kit) (Invitrogen, Waltham, MA, USA). cDNA was made using the High-Capacity cDNA Reverse Transcription Kit (#4368814; Applied Biosystems, Waltham, MA, USA).

PCR screening for SARS-CoV-2

In-hospital PCR testing was done using the Syntol kits (Moscow, Russia) on the first nasopharyngeal swab, targeting the SARS-CoV-2 (orf1ab) locus. Repeat SARS-CoV-2 real-time RT-PCR testing was performed on the second swab in the research lab using the TaqPathCov19 (Thermo Fisher Scientific, Waltham, MA, USA) targeting the SARS-CoV-2 ORF, N and S loci, following the manufacturer’s protocol.

Multiplex PCR for respiratory viruses

A multiplex PCR panel Amplisens ARVI-screen-FRT (cat#R-V57-100-F; Amplisens, Moscow, Russia) was used to test for the presence of common RNA viruses, including respiratory syncytial virus (RSV), metapneumovirus (MPV), human parainfluenza virus-1-4 (hPIV), coronaviruses (CoV) ОС43/HKU-1 and NL-63/229E and rhinovirus (hRV), and two DNA virus sub-groups- adenovirus (Adv) B, C and E and bocavirus (BoV) (see Table S1 for channel layout); the assay positivity threshold was set as recommended by the manufacturer at Ct = 32. In addition, IAV testing was conducted using the primers and protocol developed by Spackman et al. (2002) using the Qiagen one-step RT-PCR kit. All PCR reactions were run on a QuantStudio 5 Real-time PCR machine (Thermo Fisher, Waltham, MA, USA). The thresholds of detection for the PCR assays were 1 * 103 copy/ml for RSV, MPV, hPIV, BoV and hRV, 1 * 104 copy/ml for hCoV, and 5 * 103 copy/ml for Adv, while the sensitivity of IAV PCR was 10−1 EID50 of virus (Spackman et al., 2002).

Sample enrichment for virome sequencing

Sample preparation and sequencing was done following protocols optimized for virome characterization using the Ion Torrent platform (Thermo Fisher Scientific, Waltham, MA, USA) (Vibin et al., 2018; Conceição-Neto et al., 2015). Prior to library preparation, mucosal samples were purified and enriched to remove host cells and free nucleic acids as follows. Samples were first low-speed centrifuged at 5,200 g for 10 min to remove cell debris, then filtered using centrifugation through a 0.45 µm cellulose acetate filter (Spin-X Centrifuge tube filter, cat#8162; Corning Inc., Corning, NY, USA) at 13,000 g for 1 min. Subsequently, to remove free nucleic acids, the filtrate was treated with Pierce Universal Nuclease (cat#88702; Thermo Fisher Scientific, Waltham, MA, USA) at the rate of 10 units of enzyme per 1 ml of solution. Finally, samples were concentrated using a Pierce™ PES protein concentrator (cat#88503), allowing concentration of proteins at 100 K molecular weight cut-off (Thermo Fisher Scientific, Waltham, MA, USA) at 13,000 g to a final volume of 150 µl/sample.

Library preparation and sequencing

Extracted nucleic acids were subjected to cDNA synthesis, amplification and primer removal using SeqPlex RNA Amplification Kit (Sigma, St. Louis, MO, USA) according to the kit instructions to produce 150–400 nucleotide-long cDNA for downstream library construction. The quantity and quality of the amplified product were checked using the NanoDrop and Qubit, as specified above. Libraries were prepared using the Ion Plus Fragment Library Kit (Thermo Fisher Scientific, Waltham, MA, USA), Ion Xpress Barcode Adapters 1–96 Kit (Thermo Fisher Scientific, Waltham, MA, USA), Agencourt AMPure XP kit (Beckman Coulter, Brea, CA, USA). Briefly, 100 ng of the cleaned product was used for end repair using the end repair enzyme in the Ion Plus Fragment Library Kit (cat#А28950). The sample was then purified using Agencourt AMPure XP kit (cat#A63881) at a 1.8X reagent to sample ratio and eluted in 25 µL of Low TE buffer. Following elution, barcoded libraries were prepared using Ion Plus Fragment Library Kit and Ion Xpress Barcode Adapters 1–96 Kit (cat#4471250) as per the kit instructions. Quantification of unamplified libraries was performed as per the protocol from Ion Library TaqMan™ Quantitation Kit (cat#4468802) using the QuantStudio 5 Real-Time PCR System. Based on the concentration estimated, the dilution that resulted in a concentration of ~100 pM was then made. Libraries were pooled prior to loading onto Ion 530 Chips (cat#A27764) using the Ion Chef Instrument. Following template preparation, the chips were run on the Ion Torrent S5 System (Thermo Fisher Scientific, Waltham, MA, USA) following the manufacturer protocols.

Bioinformatic analyses

FASTQ-formatted reads were processed in EDGE (Li et al., 2017) following Bartlow et al. (2022) methodology. Reads were pre-processed at a minimum length = 50 bp and trim quality level = 20. Host nucleic acids were removed using the human GRCh38 reference genome with a 90% similarity cut-off. To detect reads with a similarity to known pathogens, we used read-based taxonomic classification tools Genomic Origin Through Taxonomic CHAllenge (GOTTCHA2)(Freitas et al., 2015), Kraken2 (Wood & Salzberg, 2014), Burrows-Wheeler Alignment tool (BWA) (Li & Durbin, 2009), MethaPhlAn2 (Truong et al., 2015) and Diamond (Buchfink, Xie & Huson, 2015). Phylogenetic and molecular evolutionary analyses were conducted using MEGA version 11 (Tamura, Stecher & Kumar, 2021). To reconstruct the phylogenetic relationships of the viral isolates, we created a tree using sequences from the National Center for Biotechnology Information (NCBI) satisfying one or all of the following criteria: (i) proximity to the Kazakhstan sequences, as determined by the pairwise distance in BLASTN and selecting global strains closest to the Kazakhstan isolates (n = 10 for hRV, n = 5 for hPIV-3); (ii) randomly selected full-length viral sequences from more distant sub-groups (n = 25 for hRV, n = 20 for hPIV).

Results

A total of 1,812 adults presented with upper respiratory symptoms at the outpatient departments of two polyclinics in May–June 2021, of whom 21 (1.16%) had a PCR-confirmed COVID-19 diagnosis. We then repeated SARS-CoV-2 PCR in a subset of 50 symptomatic participants with a negative COVID-19 test result. The median age of these participants was 31 years, and most were women (76%, Table 1). The most frequently observed symptoms among these participants were nasal congestion, fever, and sore throat (Table 1). Our repeat PCR testing confirmed the absence of SARS-CoV-2 within this participant subset.

Table 1 Demographic and clinical characteristics of the study participants assessed for the presence of non-COVID-19 RTI (n = 50).

Age (IQR), years	31.0 (27.0–36.0)	
Female sex, n (%)	38 (76.0)	
Respiratory symptoms	
Fever, n (%)*	34 (68.0)	
Nasal congestion with or without rhinorrhoea, n (%)	46 (92.0)	
Cough, n (%)	32 (64.0)	
Sore throat, n (%)	34 (68.0)	
Lymphadenopathy, n (%)	18 (36.0)	
Notes:

* Fever was defined as axillary temperature ≥36.7 as measured at admission.

IQR, interquartile range.

We then tested the SARS-CoV-2-negative participant subset for the presence of common RTI viruses and influenza A. Viral mono- or co-infection was detected by multiplex PCR in 33/50 (66%) subjects (Table 2 and Fig. S1). Specifically, in 25/50 (25/50, 50%) PCR identified a mono-infection with human parainfluenza virus (hPIV), most of which (24/25, 96%) was sub-typed as hPIV-4, and only one sample was hPIV-3/respirovirus-3+. Two subjects (2/25, 4%) had an hRV mono-infection and in four (4/25, 8%) samples we detected an hPIV4-hRV co-infection. Two subjects (4%) had either adenoviral or OCR43/HKU-1 coronavirus mono-infection. No IAV, RSV, MPV, BoV, or CoV NL-63/229E was detected.

Table 2 Respiratory viruses detected by multiplex PCR and sequencing.

Results for 33 (out 50) symptomatic participants whose PCR findings were positive on the multiplex RTI or IAV PCR.

Sample	PCR result	Ct score	Sequencing result	Number of reads (linear coverage, %)**	
7	Adv	27.94			
9	hCov (HKU-1/OC43)	24.7			
13	hPIV-4	28.6			
14	hRv	20.05	HRV-A56	19; 7 (36.5)	
43	hPIV-4	28.16			
48	hPIV-4	29.05			
50	hPIV-4	26.9			
52	hPIV-4	28.46			
53	hPIV-4	26.4			
54	hPIV-4	24.4			
55	hPIV-4	22.65			
59	hPIV-4	23.04			
66	hPIV-4	24.5			
73	hPIV-4	26.2			
74	hPIV-4	27.13			
75	hPIV-4, hRV	26.3, 16*	HRV-A30	6; 1 (30.3)	
77	hPIV-4	24.2			
78	hPIV-4, hRV	24.86, 22	HRV-A100	25; 8 (39.6)	
79	hPIV-4	23.33			
80	hPIV-4	22.64			
81	hPIV-4	24.78			
83	hPIV-4	23.55			
84	hPIV-4	23.17			
85	hPIV-4	21.8			
87	hPIV-4, hRV	21.9, 16	HRV-A1B	11,155; 69,393 (100)	
91	hPIV-4	27.28			
96	hPIV-4	28.58			
99	hPIV-4	25.85			
109	hPIV-4, hRV	31.7, 31	HRV-A1B	−; 2 (4.8)	
112	hPIV-4	27.87			
123	hPIV-4	28.47			
127	hPIV-4	28.97			
12	hPIV-3	21.9	HPIV-3	256; 337 (69.1)	
Notes:

* For co-infections Ct values are given for HPIV and HRV, respectively.

** Number of sequencing reads was estimated by BWA (given first) and Kraken2 (given second) approaches.

Subsequently, we performed virome sequencing of all PCR+ samples and were able to characterize viral sequence diversity in 6/33 (18%) samples, consisting of 5 hRV+ and 1 hPIV-3+ (Table 1). All hRV were classified as species A (hRV-A) represented by four sub-strains: A56, A30, A100 and A1B (Table 1 and Fig. 2), while the single hPIV-3 (respirovirus-3) clustered distinctly with hPIV-3, but not other hPIV sequences (Fig. 3).

Figure 2 Reconstruction of the phylogenetic relationships for the detected rhinovirus (probes #14, 78, 75, 87 and 109).

The consensus sequences obtained using whole virome sequencing were aligned with a select subset of complete virus genomes (see Methods) using the ClustalW algorithm in MEGA 11. The phylogenetic tree was reconstructed using the maximum likelihood method and bootstrap 500.

Figure 3 Reconstruction of the phylogenetic relationships for the detected respirovirus-3 (probe #12).

The consensus sequences obtained using whole virome sequencing were aligned with a select subset of complete virus genomes (see Methods) using the ClustalW algorithm in MEGA 11. The phylogenetic tree was reconstructed using the maximum likelihood method and bootstrap 500.

Discussion

Here we report data on the prevalence of viruses associated with symptomatic upper RTI in a cohort of adult outpatients in Almaty, Kazakhstan during a period of low SARS-CoV-2 transmission in Spring 2021. The prevalence of laboratory-confirmed COVID-19 in the cohort was under 2%, consistent with low nation-wide SARS-CoV-2 circulation as reported by the government surveillance platforms for the study period (https://www.coronavirus2020.kz/, accessed February 1 2023). Most RTI in the cohort (>60%) were associated with hPIV and hRV as determined by multiplex PCR on par with recent data from other regions, where symptomatic RTI have been associated with common non-coronaviral pathogens (Wan et al., 2021; Varela et al., 2022; Hyams et al., 2022; Olsen et al., 2021). Sequencing of the PCR+ samples allowed to reconstruct the diversity of the isolated hRV and hPIV-3/human respirovirus-3.

Our multiplex PCR assay identified viral pathogens in 66% of the tested subset. This is consistent with other studies reporting a similar PCR sensitivity in cohorts of symptomatic adults and children (Varela et al., 2022; Bartlow et al., 2022) . We speculate that in our cohort some infections were undetectable due to the limited nature of our PCR panel, which, while focusing on most prevalent community RTI and IAV, did not target some viral and bacterial infections, such as other subtypes of influenza or Bordetella pertussis. Other factors that could have impacted the sensitivity of our PCR analysis are low viral loads and sample loss during sample collection and laboratory manipulation.

RTI are some of the most common causes of human illness worldwide (Jin et al., 2021), and most RTI have viral aetiology. Over the last century, RTI prevention efforts have had a limited focus on reducing the burden of influenza, whooping cough, and more recently COVID-19, while infections by pathogens such as parainfluenza or rhinovirus remain poorly addressed, despite causing substantial morbidity and mortality globally but especially in less developed countries with sub-optimal access to healthcare and nutrition among other factors (Jin et al., 2021; Jacobs et al., 2013; Branche & Falsey, 2016; Yegorov et al., 2019). Such limited understanding of RTI has implications for healthcare resource allocation and is especially important for economically challenged regions such as Kazakhstan, where outbreaks of non-COVID RTI continue to strain the healthcare system resources.

Kazakhstan shares borders with several states of Central Asia, Russia, and China, which in combination with a well-developed transit network could enhance the risk of pathogen importation and rapid community spread. In keeping with this, our phylogenetic analyses here indicate that the hRV-A circulating in Kazakhstan is highly diverse, with some species having global origins, suggesting that international travel plays an important role in the dissemination of community RTI. Although our study was underpowered to conduct a formal analysis of phylogenetic transmission clusters, we noted the presence of a single cluster consisting of two subjects with hRV-A1B infection. These subjects were both female and appeared at the outpatient department within a span of 5 days, which suggests a shared origin for infection in these participants. However, we were unable to gather any additional evidence in support of this.

Our findings should be interpreted in the light of the limitations. First, our RTI analysis was conducted on a small participant subset, which may have biased total proportions of RTI pathogens detectable among the outpatients. Next, the cross-sectional nature of the analysis prohibits the analysis of seasonality and changes in pathogen dynamics that may have occurred over time. We were able to sub-type RTI pathogens by sequencing only in a small number of samples. However, this is not surprising given that PCR is more sensitive than sequencing due to its pathogen-targeted nature. Importantly, pathogen detection does not prove a causal relationship with symptomatology. Despite the limitations, our study provides important insight into RTI epidemiology during the low COVID-19 transmission period in Kazakhstan, adding to the growing knowledge on health conditions for the region (Yegorov et al., 2021; Kadyrova et al., 2022; Yegorov et al., 2020).

In summary, to the best of our knowledge, our study is the first to describe RTI dynamics in a population from Central Asia and Kazakhstan, providing pilot data for the future, more powered RTI surveillance studies in the region.

Supplemental Information

Supplemental Information 1 Supplementary Tables.

Click here for additional data file.

We thank all the study participants and the clinic staff.

Additional Information and Declarations

Competing Interests

Author Contributions

Human Ethics

DNA Deposition

Data Availability

Vyacheslav Yu. Beloussov & Joanna Granica are employed by TreeGene Molecular Genetics Laboratory.

Nurlan Sandybayev conceived and designed the experiments, analyzed the data, prepared figures and/or tables, and approved the final draft.

Vyacheslav Beloussov conceived and designed the experiments, analyzed the data, authored or reviewed drafts of the article, and approved the final draft.

Vitaliy Strochkov performed the experiments, analyzed the data, prepared figures and/or tables, and approved the final draft.

Maxim Solomadin performed the experiments, authored or reviewed drafts of the article, and approved the final draft.

Joanna Granica performed the experiments, prepared figures and/or tables, and approved the final draft.

Sergey Yegorov conceived and designed the experiments, performed the experiments, analyzed the data, prepared figures and/or tables, authored or reviewed drafts of the article, and approved the final draft.

The following information was supplied relating to ethical approvals (i.e., approving body and any reference numbers):

All study procedures were approved by the Commission on bioethics of KazNARU (dated October 15, 2020). Written informed consent was obtained from all participants.

The following information was supplied regarding the deposition of DNA sequences:

The sequences are available at GenBank: SRX18371932-SRX18371937.

The following information was supplied regarding data availability:

The sequence reads are available at NCBI: PRJNA904925.

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
