# Peer review of "Characterization of viral pathogens associated with symptomatic upper respiratory tract infection in adults during a low COVID-19 transmission period"

_PeerJ, doi:10.7717/peerj.15008_

## Round 0.1 · original submission · Major Revisions

Please address the concerns of both reviewers and revise the manuscript accordingly.

Reviewer 1 ·

Basic reporting

The manuscript is written clearly and english is good. The authors included all relevant references. However more data is needed to support hypotheses.

Experimental design

Experimental design could be improved upon.

Validity of the findings

Findings are good but not enough to support authors claim.

Additional comments

1. Please provide identification number/ catalogue number of all kits used.
2. In figure 2 and figure 3 legend please change from Clastal W to ClustalW.
3. In table 1, please show PCR results as plots.
4. Please make a table showing clinical characteristics of individuals.
5. Include more data like distribution of respiratory virus types in patients and phylogenetic transmission clusters
6. In addition to PCR result in table, include data of viral load.
7. Since this study is location specific, include data of seasonality of infections and meteorological profiles.

·

Basic reporting

In This work, Sandybayev and colleagues have presented the data to provide information regarding prevalence of non-COVID RTI in Kazakhstan during the period of low COVID transmission. I am appreciating the concept of the study. In Methodology, Authors provided the technical reasoning for each technique they have used. I found this article is very informative to public health guidelines for management of RTI.

Experimental design

1. The authors as they have mentioned the study is performed with very limitations which includes the crucial points for the study. Mostly the no.of patients for the study which is the most important criteria to delineate the prevalence or dynamic of RTI. Authors failed to explain why they have the patient number limited to 50 out of 1791 patients with COVID negative PCR results (Lane 169).

Validity of the findings

1. As this study depends on the dynamics of non-COVID-RTI infections more background literature is required about the prevalence of the Viruses.
Minor:
2. Lane 115: Catalogue number for Multiplex PCR kit is missing.
3. Results and Discussion: Authors did not mention about the Primers and probe sets used in Multiplex kit. If they mention that it will be helpful to see the results
4. Multiplex Detection limits of the test are not mentioned.
5. In this study, Authors have identified the other viruses which cause RTI.
6. Line 203-208: As authors have the only information (data) for other RTI Viruses like HRV-A and HPIV during the certain period of time so it’s not really valid to discuss the points mentioned.
7. Table 1: Why there are only HPiv and HRV ct values. Authors did not mention other viruse values described in lane 115-120.

Additional comments

1. Draft needs to be arranged properly.
2. More References are required in the entire draft mainly in the results and discussion section.

---

## Round 0.2 · accepted · Accept

All critiques were addressed, and the revised manuscript is acceptable now.

Reviewer 1 ·

Basic reporting

The authors have satisfactorily responded to all comments, the article can now be accepted for publication.

Experimental design

NA

Validity of the findings

NA

Additional comments

NA